# Impact of COVID-19 on Physical Activity and Lifestyles in Post-Confinement Sports Science Undergraduates

**DOI:** 10.3390/ijerph19159115

**Published:** 2022-07-26

**Authors:** Ismael García-Campanario, Luc E. Vanlinthout, Rocío Toro, Alipio Mangas, Carolina Lagares-Franco

**Affiliations:** 1Faculty of Medicine, University of Cadiz, 11003 Cadiz, Spain; rocio.toro@uca.es (R.T.); alipio.mangas@uca.es (A.M.); carolina.lagares@uca.es (C.L.-F.); 2Faculty of Medicine, University Hospital Gasthuisberg, University of Leuven, 3000 Leuven, Belgium; vanlinthout.l@skynet.be; 3Research Group INIBICA CO15: Population and Health: Determinants and Interventions, 11009 Cadiz, Spain

**Keywords:** COVID-19, university students, physical activity, lifestyle, Mediterranean diet, confinement

## Abstract

The aim of this study was to assess whether the infection by SARS-CoV-2 has significantly influenced physical activity, diet, alcohol, and drug consumption habits, as well as the quality of life of students of the bachelor’s degree in Physical Activity and Sports Sciences. For this purpose, an online survey was conducted, which included socio-demographic questions related to the COVID-19 disease. Physical activity was analyzed using the International Physical Activity Questionnaire (IPAQ), adherence to the Mediterranean diet using the PREDIMED questionnaire, alcohol consumption using the AUDIT questionnaire, and drug consumption using the DAST-10 questionnaire. Health-related quality of life was analyzed with the SF-12 questionnaire. Our results reveal that those who engaged in either vigorous physical activity or, on the contrary, very low-intensity physical activity, were affected by the SARS-CoV-2 disease, which reduced the average weekly time they spent on their type of activity. However, those who previously performed moderate activities have managed to stay on the same fitness level despite having suffered from SARS-CoV-2 disease (*p* = 0.433). In conclusion, general health is affected by suffering from the COVID-19 disease, inadequate eating habits, substance use, and the performance of vigorous or very low-intensity of physical activity.

## 1. Introduction

On 14 March 2020, the Government of Spain declared a State of Alarm throughout the country in order to deal with the health emergency caused by COVID-19. In efforts to reduce the spread of the virus, social restrictions have been implemented. These included social distancing, constraints on traveling, and restricted access to sports facilities, health centers, universities, and work and leisure places. Social isolation and loneliness, during lockdowns, are associated with increased physical inactivity and sedentary time, weight gain, addiction, behavioral disorders, as well as with insufficient exposure to sunlight [1]. Imposed social restrictions with lack of interaction and emotional support were determinants related to mental health conditions in the young population [2]. Other factors such as boredom, misinformation, frustration, supply concerns, and financial loss could trigger mood changes and deterioration of mental health [3].

The COVID-19 pandemic aggravated the occurrence of symptoms of anxiety, depression, panic, distress, fear, and stress response in the general population [4]. Mandatory confinement has caused a great impact on physical activity, dietary habits, and mental health, especially in a healthy population without a history of mental disorders and illnesses [5]. Among university students, stress from academic activities is already a factor that negatively affects their health [6]. Although, students of the bachelor’s degree in Physical Activity and Sports Sciences in Spain have many hours of practice, the level of physical activity among university students is low [7].

People in lockdown are more prone to a sedentary lifestyle and unhealthy eating habits with reduced fruit and vegetable intake. Furthermore, stress, depression, and anxiety act as triggers for alcohol and tobacco use as a way of coping with painful emotions [1,2,3,4]. The Mediterranean diet is able to preserve cardio-metabolic health by improving lipid profile, and insulin resistance. Furthermore, it can protect against arterial hypertension, stroke, and cancer [5,6,7,8]. Students adhering to the Mediterranean diet spent more time on physical activity, whereas those with unhealthy eating patterns appeared to be unable to improve their dietary behavior [9].

It is our aim to explore the changes in lifestyle habits and in physical and emotional health once students have returned to traditional classroom learning. More specifically, we wanted to assess how healthy and unhealthy behaviors during the lockdown period impacted the post-confinement quality of life. This was performed on a sample of university students of the bachelor’s degree of Physical Activity and Sports Sciences at the University of Cádiz.

## 2. Materials and Methods

This is a descriptive cross-sectional study involving young people over 18 years of age from the bachelor’s degree in Physical Activity and Sports Sciences of the University of Cadiz (UCA). On average, about 300 students study this degree each year at this university. It is estimated that 60% of university students were insufficiently active during the COVID-19 pandemic [10]. Sample size calculation was performed using the formula of Cochran [11]. Taking this percentage as a reference, the number of students enrolled and assuming a confidence level of 95% and an estimation error of 5%, a minimum of 166 students will be necessary for this study.

An online survey was conducted that included socio-demographic information such as age, sex, the student’s academic year of enrollment, i.e., 1st, 2nd, 3rd, or 4th year and body mass index (BMI). With respect to COVID-19, the following items were recorded: the student’s vaccination status (vaccinated/unvaccinated), whether he/she had suffered from the disease in the last 6 months (yes/no), or if he/she had needed hospitalization (yes/no). Furthermore, we inquired into the occurrence of common symptoms such as fever, cough, loss of taste, loss of smell, respiratory distress, etc.

The International Physical Activity Questionnaire (IPAQ), an instrument validated in a university population [8], was used to determine the level of physical activity. The IPAQ is composed of 7 items that assess the frequency (days per week), duration (time per day), and intensity (light, moderate or vigorous) of physical activity performed during the last week.

Three questionnaires were used to obtain information related to consumption habits: the PREDIMED (Prevention with Mediterranean Diet) questionnaire; the Alcohol Use Disorders Identification Test (AUDIT) and the Drug Abuse Screening Test (DAST-10) for substance abuse. The PREDIMED questionnaire is composed of 14 dichotomous items (scores yes/no or 0/1), assessing adherence to the Mediterranean diet. A total score below 9 points implies the need to improve the quality of the diet. This instrument has been validated in the university population [9]. The AUDIT test is composed of 10 dichotomous items and has been validated in the university population [12]. A total value of 8 or more in men and 6 or more in women indicates excessive alcohol consumption. The DAST-10 questionnaire [13] is an instrument validated in the Spanish population [14] that records drug use through 10 dichotomous items. An overall score of 0 indicates no health risk; from 1–2 mild risk; from 3–5 moderate risk; and from 6 points onwards serious risk.

Finally, the 12-item Short Form Survey (SF-12) is a self-reported outcome measure assessing the impact of health on an individual’s everyday life. It is often used as a degree of well-being and functional capacity measure of people over 14 years of age [15]. The SF-12 uses eight domains: (I) General health (GH) assessed with item 1, which includes the personal assessment of health; (II) physical functioning (PF) (items 2 and 3) which evaluates the degree to which health limits physical activities such as climbing stairs or walking for more than one hour; (III) role-physical (RP) (items 4 and 5) determine the degree to which physical health affects work and other daily activities; (IV) role-emotional (RE) (items 6 and 7) appraise the degree to which emotional problems affect work or daily activities is assessed; (V) bodily pain (BP) (item 8) estimates its effect on work and home; (VI) mental health (MH) (items 9 and 11) gauge feelings of calmness, peacefulness, discouragement, and sadness; (VII) energy/fatigue (VT) (item 10) measures the degree of vitality versus that of fatigue and exhaustion; and (VIII) social function (SF) (item 12) rates the degree to which physical or emotional health problems interfere with daily life.

The information on each item is collected through a Likert scale with scores ranging from 1 to 6 points, resulting in a total score between 12 and 47. The higher the score, the better the health-related quality of life. This tool has been validated in the general population [16,17,18].

The data were collected using a Google Forms form and exported to Excel where the quality control of the information and the respective coding was carried out. The form was sent to the students through the class delegates. Participation was completely voluntary, and participants could withdraw from the study at any time without having to give a reason for withdrawal and without consequences. The study was conducted in accordance with the Helsinki declaration and was approved by the Bioethics Committee of the University of Cadiz.

### Statistical Analysis

A descriptive analysis of the data from the survey was carried out using frequencies and percentages for the qualitative variables and means and standard deviations (SD) for the quantitative variables. Subsequently, different hypothesis tests were carried out, based on the partial and final scores in each of the questionnaires used. After confirming normality using the Kolmogorov–Smirnov test, Shapiro–Wilk test, or Q–Q plot, we used Student’s *t*-test. If the normality assumption was not satisfied, non-parametric tests were applied to equivalent non-parametric tests such as the Mann–Whitney U-test or the Wilcoxon rank-sum test. A significance level of 5% was assumed and the IBM SPSS v.26 program was used.

## 3. Results

A total number of 168 participants was included, mainly males (76.8%) with an average age of 21.19 (3.31) years. About one-third of the participants (34.9% of men and 33.3% of women) claimed to have suffered from a COVID-19 infection (Table 1). None of them needed to be hospitalized. Moreover, 74.6% had the complete vaccination schedule and only 2.4% were not vaccinated, the rest (23.0%) were waiting for a second dose.

Comparisons were made between participants that have gone through the COVID-19 infection with those who have not. No statistically significant differences could be demonstrated in alcohol or drug consumption or in eating habits (Table 2). Further analysis revealed that item 11 of the PREDIMED questionnaire was significantly different (*p* < 0.001), indicating that the weekly consumption of commercial (non-homemade) pastries such as cookies, custards, sweets, or cakes was higher among those who had suffered from the disease.

Table 3 compares the results of the International Physical Activity Questionnaires (IPAQ) between participants who have gone through a COVID-19 infection and those who have not. Post-COVID-19 participants who engaged in either vigorous or low-intensity physical exercise managed to maintain the number of training days (*p* = 0.045) but reduced the average daily time they spent on these activities (*p* = 0.010). Only those who performed moderate physical activity kept their exercise routine unchanged after having gone through COVID-19 disease (*p* = 0.317 [days] and *p* = 0.433 [time], respectively).

Table 4 summarizes the eight dimensions of the SF-12 questionnaire with respect to sex, adherence to the Mediterranean diet, physical activity, consumption of alcohol and drugs, and the post-COVID-19 condition. General health in these students was found to be compromised by their consumption habits. Participants with insufficient adherence to the Mediterranean diet or those suffering from the consequences of alcohol and drug consumption had an inferior perception of their state of health. Female sex, and undesirable alcohol and drug consumption were associated with inferior results for moderate physical activity, such as walking for more than one hour. Noxious habits of drug consumption and a post-COVID-19 condition yielded lower performance in more strenuous exercise, such as climbing several floors upstairs. After having suffered from COVID-19 disease the role-physical was negatively affected, with the consumption of drugs and lower levels of exercise. The outcome of participants who had carried out vigorous physical activities was better than that of those performing physical activities of low or moderate intensity.

Adherence to the Mediterranean diet had a beneficial effect on general mental health. Furthermore, mental health, energy/fatigue, and adherence to the Mediterranean diet were found to be negatively associated with the consumption of drugs. Finally, social functioning was shown to be compromised by excessive consumption of alcohol and drugs.

## 4. Discussion

The current study evaluates the health status of students of the bachelor’s degree in Physical Activity and Sports Sciences after the end of the confinement, thereby assessing the extent to which they have been affected by the COVID-19 disease. The main finding was that suffering from the disease has no disadvantage for students who performed moderate physical activity. However, those who engaged in either vigorous physical activity or, on the contrary, very low-intensity physical activity, were affected by the disease, which reduced the average weekly time they spent on this type of activity.

### 4.1. COVID-19 and Alcohol and Drug Use

Alcohol affects many organs. Even in moderate amounts, alcohol results in a subclinical immunosuppression that becomes clinically relevant after a secondary insult (e.g., bacterial or viral infection). Moreover, there is a dose-dependent impact [19].

Researchers have raised concerns that a pandemic may increase the risk of excessive alcohol consumption. COVID-19 and unhealthy drinking patterns are closely related, even so in the young population. Those with more symptoms of depression and anxiety reported greater increases in alcohol consumption as compared to students with fewer symptoms. Furthermore, students with greater perceived social support reported less alcohol consumption. A role of the college campus environment in perpetuating the risk of heavy drinking has been demonstrated. Moving off-campus or having greater parental monitoring set limits on heavy drinking behavior [20].

Numerous studies have addressed the alcohol and drug consumption among young bachelor students during the COVID-19 pandemic. A study with Dutch students found that weekly tobacco consumption remained stable, whereas excessive alcohol consumption (binge drinking) decreased, and cannabis use increased. On the other hand, a study conducted at four German universities revealed excessive alcohol consumption in 45.8%, tobacco use in 19.4%, and cannabis consumption in 10.8% [21]. These figures differ from those obtained in the current study, where 21.43% reported excessive alcohol consumption and 8.33% moderate or severe risk of drug use. These figures are more in line with those found in a study conducted in the Community of Madrid where the AUDIT questionnaire classified 26.2% of respondents as high-risk drinkers [22].

Differences regarding alcohol and substance abuse at the different universities may be explained by the student’s residency environment, i.e., the campus that more easily tolerates all these kinds of activities versus the home environment with parental monitoring [23]. Furthermore, the drug policy in some countries may bias the behaviors relative to cannabis consumption.

### 4.2. COVID-19 and Adherence to the Mediterranean Diet Athletes’ Endurance Exercise Performance

Dietary habits are closely associated with the level of physical activity. A cluster analysis revealed that athletes who submitted to higher levels of physical activity were more likely to adhere to healthy eating patterns, regardless of quarantine. The impact of this pandemic situation on lifestyle has been widely reported [24,25,26]. Psychological distress and solitary confinement are the main reasons for the high consumption of processed foods. Food with high content of saturated fats, refined carbohydrates and sugars, and low levels of fiber, unsaturated fats, and antioxidants have replaced the Mediterranean diet during the episode of lockdown. The Mediterranean diet, which is rich in macro and micronutrients without any food contaminants, has shown to be amongst the healthiest in the world [27].

A study conducted on 49 student-athletes showed a medium-low adherence to the Mediterranean diet. Their nutritional standards were unrelated to the regular physical activity they performed [28]. Some social groups such as women and the elderly, however, managed to maintain their Mediterranean eating habits during the pandemic period [23,29,30]. Furthermore, students of Health Sciences showed a similar attitude [31] with a high degree of adherence to the Mediterranean diet in 58.6%, an intermediate degree in 38.6%, and poor adherence in 5.0%. This healthy behavior, probably, was inspired by the knowledge that nutrition has a key role in the protection against viral aggressions. On the contrary, Lopez-Moreno et al. [22] reported poor adherence to the Mediterranean diet in 63.6% of participants, with no difference between the sexes. Our results agree with those obtained by the study of Lopez-Moreno et al. in which 67.07% of participants needed to improve adherence to the Mediterranean diet. The eating habits of participants of the current study, which were unrelated to their sex, changed after having gone through a COVID-19 infection with a higher proportion of industrial pastries consumed by those who had suffered from the disease. This behavior seems paradoxical and is in disagreement with health logic.

The Mediterranean diet is a primarily plant-based eating plan that includes whole grains, olive oil, fruits, vegetables, legumes, and nuts. Certain of its nutrients reduce the levels of oxidized LDL and decrease the expression of oxidative stress- and inflammation-related genes [32]. The expression of these genes is a major event that can modify the immune response. Therefore, a diet rich in saturated fatty acids can trigger the immunomodulatory response by promoting the inflammatory surge [33]. Although COVID-19 infection cannot be prevented by any food or dietary supplements, maintaining a healthy diet is an important part of supporting a strong immune system. Demotivation, psychological distress, and anxiety can explain an increased consumption of meals, snacks, and processed food. Although students had more spare time, they continued to resort to industrial processed fast food.

### 4.3. COVID-19 and Physical Activity

Physical activity has significant health benefits at several levels [34]. The practice of physical activity before, during, and after confinement has undergone modifications among university students. Social and physical distancing measures, which have become commonplace to curtail the spread of the disease, have disrupted many regular aspects of life, including sports and physical activity. Training centers and sports clubs were closed, team sports were suspended, and regular training routines of athletes were discontinued. Despite WHO recommendations for continued physical activity [35], the severity of the restrictions during the lockdown in Spain was a major challenge for maintaining mental and physical health. Some studies associated confinement with an increase in sedentary lifestyle and unhealthy eating behaviors [36,37]. Engagement in physical activity declined because of the requirement to self-isolate and to stay in place. Despite access to various online platforms, remotely delivering exercise classes and training programs, social isolation, and loneliness negatively impacted sedentarism [24].

It has been shown, however, that an active lifestyle prior to the lockdown favored the practice of adequate physical activity during confinement. Moreover, open-room availability or scheduled activities may have facilitated additional opportunities to engage in physical activity [38,39,40]. One of the few studies, that assessed changes in physical activity among students after the COVID-19 pandemic using the IPAQ questionnaire, observed that, although the level of physical activity during the lockdown decreased, exercise behavior increased by 50% of respondents after exiting from lockdown [41]. Our population represents a very active sample as far as physical activity is concerned. Part of our study population has seen this lifestyle change after going through the disease. The only post-COVID-19 participants who kept their active lifestyle unchanged were those engaged in moderate physical activity. Although controversial, it has been reported that the performance of moderate physical activity can protect against viral or bacterial infections by enhancing immune function [42,43]. Some studies support the idea that regular exercise training and higher levels of physical activity and fitness have an overall anti-inflammatory and immune-boosting effect mediated through multiple pathways [44]. Other studies, however, reported that high-intensity exercise does not lead to any clinically relevant immune response. High training workloads and the associated physiological, metabolic, and psychological stress associated with competition events could eventually result in transient immune perturbations, inflammation, oxidative stress, muscle damage, and vulnerability toward illnesses [45].

### 4.4. COVID-19 and Health-Related Quality of Life (SF-12)

The self-perceived quality of life among university students has been altered by the COVID-19 pandemic in terms of attitude and aptitude towards habitual daily activities [46,47,48,49]. Other studies revealed that, regardless of the level of physical activity, COVID-19 and quarantine significantly decreased the quality of life of both male and female students [50]. Students who did not engage in physical activity during confinement reported a lower level of mental health [51]. Furthermore, tobacco and cannabis use was found to be associated with excessive alcohol consumption among students [52]. The more negative the perception of psychological well-being, the greater the tendency to use tobacco and alcohol [53]. These results are consistent with the findings of the current study, in which the different dimensions of the SF12 questionnaire were affected by the eating pattern, alcohol and drug consumption, but not by gender, physical activity, or COVID-19 status.

### 4.5. Limitations

The most relevant limitation of the current investigation is the lack of similar studies assessing the physical and emotional status of university students after relaxing the COVID-19 restrictions and returning to classroom attendance of the courses. Such studies should take into account whether or not participants have suffered from COVID-19 disease. Therefore, it is not possible to compare our results with previous research in this area. Another limitation is the subset (university degree at a specific university) of the target population from which the sample is actually selected. Because of practicalities, we decided to recruit only students of the bachelor’s degree in Physical Activity and Sports Sciences of the University of Cadiz. This can be considered a weakness as well as a strength. The inclusion criteria of the current study place constraints on the ability to generalize from the results and to further extrapolate the conclusions to the entire population/society. Students of the bachelor’s degree in Physical Activity and Sports Sciences all over the world might be considered as the target population from which the participants of the current study represent a relevant sample. The current study, therefore, allows drawing conclusions about these students of the bachelor’s degree in Physical Activity and Sports Sciences.

## 5. Conclusions

This paper investigated the impact of COVID-19 on physical activity and lifestyles in post-confinement sports science undergraduates. The main conclusion of this study is that performing moderate physical activity allows individuals, once they have passed the COVID-19 disease, to return to their daily needs and obligations more normally. The psychological stress and anxiety after suffering COVID-19 impacted negatively this cohort with low adherence to the Mediterranean diet. Their perceived general health status is affected by the COVID-19 disease, inadequate nutritional habits, toxic substance abuse, and a trend toward a sedentary lifestyle.

## Figures and Tables

**Table 1 ijerph-19-09115-t001:** Descriptive statistics.

		Total	Men	Women
		*N* (%)	*N* (%)	*N* (%)
Course	1st	71 (42.5)	47 (66.2)	24 (33.8)
	2nd	26 (15.6)	26 (100)	0 (0)
	3rd	41 (24.6)	32 (78)	9 (22)
	4th	29 (17.4)	23 (79.3)	6 (20.7)
BMI	Underweight (<18.5)	4 (2.4)	2 (50)	2 (50)
	Normal weight (18.5–24.9)	129 (77.2)	93 (72.1)	36 (27.9)
	Overweight (25–29.9)	29 (17.4)	29 (100)	0 (0)
	Moderate obesity (30–34.9)	3 (1.8)	3 (100)	0 (0)
	Severe obesity (>35)	2 (1.2)	2 (100)	0 (0)
Vaccinated	No	4 (2.4)	4 (100)	0 (0)
	Yes	126 (75.0)	92 (73)	34 (27)
	Yes, missing dose	38 (22.6)	33 (86.8)	5 (13.2)
Have you been infected with COVID-19 in the last 6 months?	Yes	58 (34.5)	45 (77.6)	13 (22.4)
	No	110 (65.5)	84 (76.4)	26 (23.6)

**Table 2 ijerph-19-09115-t002:** Global scores of the IPAQ, PREDIMED, AUDIT, DAST, and SF12 questionnaires depending on the COVID-19 condition.

		COVID-19		
		YES	NO	*p*-Value
IPAQ		3762.41 (2142.63) *n* = 55	5017.66 (3152.81) *n* = 103	0.004
PREDIMED	You need to improve your diet	42 (38.2%)	68 (61.8%)	
	Good quality diet	14 (25.9%)	40 (74.1%)	0.083 *
AUDIT	No excessive consumption	45 (34.1%)	87 (65.9%)	
	Excessive consumption	13 (36.1%)	23 (63.9%)	0.484
DAST10	No risk	33 (45.2%)	40 (54.8%)	0.666
	Mild risk	14 (37.8%)	23(62.2%)	
	Moderate risk	2 (25%)	6 (75%)	
	Severe risk	1 (50%)	1 (50%)	
SF12		30.65 (2.1) *n* = 54	31.02 (2.25) *n* = 107	0.315

* Unilateral significance.

**Table 3 ijerph-19-09115-t003:** IPAQ questionnaire according to COVID-19 condition.

Item	COVID-19	*n*	Mean (sd)	Sig.
Number of days a week in which he performed intense physical activity (football, swimming, heavy lifting, …).	No	110	3.51 (1.8)	0.610
	Yes	58	3.36 (1.71)	
Total time in minutes spent on such intense activity	No	109	87.34 (55.28)	0.045
	Yes	58	70.78 (39.61)	
Number of days a week in which he/she performed moderate physical activity (cycling, dancing, playing non-professional tennis, …).	No	110	2.66 (2.17)	0.317
	Yes	57	2.35 (1.76)	
Total time in minutes spent on such moderate activity	No	110	56.1 (48.41)	0.433
	Yes	57	64.56 (90.96)	
Number of days a week you walked for more than 10 min	No	110	5.28 (1.96)	0.265
	Yes	56	4.88 (2.33)	
Total time in minutes spent walking	No	104	80.43 (86.5)	0.010
	Yes	56	54.2 (40.72)	
Time spent sitting on business days for the past 7 days.	No	108	270.93 (243.05)	0.073
	Yes	57	210.8 (178.56)	

**Table 4 ijerph-19-09115-t004:** *p*-values for SF12 questionnaire by sex, COVID-19 disease and eating habits, alcohol and drug consumption, and physical activity.

SF12	SEX (Male/Female)	PREDIMED (Needs Improvement/Good Diet)	IPAQ (Low/Moderate/Vigorous)	DAST10 (No Risk/Mild Risk/Moderate Risk/Severe Risk)	AUDIT (No Excessive Consumption/Excessive Consumption)	COVID-19 (Yes/No)
Item 1 (GH *)	0.133	0.050	0.233	0.000	0.002	0.238
Item 2 (PF *)	0.015	0.586	0.960	0.000	0.023	0.125
Item 3 (PF *)	0.164	0.767	0.089	0.004	0.122	0.017
Item 4(RP *)	0.813	0.383	0.000	0.020	0.628	0.017
Item 5 (RP *)	0.621	0.363	0.822	0.065	0.800	0.003
Item 6 (RE *)	0.067	0.073	0.940	0.485	0.704	0.742
Item 7 (RE *)	0.258	0.002	0.769	0.367	0.849	0.316
Item 8 (BP *)	0.156	0.192	0.058	0.774	0.077	0.695
Item 9 (MH *)	0.131	0.067	0.295	0.019	0.160	0.475
Item 10 (VT *)	0.306	0.050	0.714	0.802	0.043	0.513
Item 11 (MH *)	0.299	0.559	0.444	0.880	0.168	0.313
Item 12 (SF *)	0.142	0.329	0.876	0.000	0.017	0.401

* GH = general health; PF = physical functioning; RP = role-physical; RE = role-emotional; BP = bodily pain; MH = mental health; VT = energy/fatigue; SF = social functioning.

## Data Availability

Not applicable.

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
