# Peer review of "Impact of COVID-19 on Physical Activity and Lifestyles in Post-Confinement Sports Science Undergraduates"

_ijerph, 2022, doi:10.3390/ijerph19159115_

Round 1

Reviewer 1 Report

The only suggestion for the authors is change from present simple to past simple the first sentence of the abstract " The aim of this study WAS...."

Author Response

Thank you. Your suggestion has been made.

Reviewer 2 Report

This paper investigated the impact of COVID19 on physical activity and lifestyles in postconfinement sports science undergraduates. The authors investigated the subjects' physical activity, diet, alcohol, and drug consumption habits using questionnaires and compared the differences between the infected and non-infected subjects. This paper is greatly weakened by the fact that there is no in-depth analysis of the reasons in the discussion section. In addition, the introduction part needs to be better organized.

Abstract

The abstract does not correctly address all sections of the work. Conclusions are needed in this part.

Introduction

The introduction mainly described the harm of COVID-19 to society and humans, which didn’t focus on the highlights of the study. The author needs to add some information about the physical activity, diet, alcohol, and drug consumption habits of the patients. In addition, the number of relative researches was large, it would be better to add some literatures to support the summary of the previous studies.

Methods

Lacking the description of the sample size, which needs to be calculated by “Cohen's d” (or others). In addition, the basic information about the subjects also needs to be added (Page 2, Line 58).

Results

In table 1, the numbers of the subjects were inconsistent. For example, the total number of students in 2nd was 27, while there were 26 male students and no female students. (There were also the same errors in BMI and Course) It’s essential to adjust the table to avoid the misunderstandings (Page 3, Line 126).

Discussion

The discussion quotes a lot of previous literatures but does not explain the reasons for the results. It is essential to add an explanation. The number of the reference was not adequate.This paper investigated the impact of COVID19 on physical activity and lifestyles in postconfinement sports science undergraduates. The authors investigated the subjects' physical activity, diet, alcohol, and drug consumption habits using questionnaires and compared the differences between the infected and non-infected subjects. This paper is greatly weakened by the fact that there is no in-depth analysis of the reasons in the discussion section. In addition, the introduction part needs to be better organized.

Abstract

The abstract does not correctly address all sections of the work. Conclusions are needed in this part.

Introduction

The introduction mainly described the harm of COVID-19 to society and humans, which didn’t focus on the highlights of the study. The author needs to add some information about the physical activity, diet, alcohol, and drug consumption habits of the patients. In addition, the number of relative researches was large, it would be better to add some literatures to support the summary of the previous studies.

Methods

Lacking the description of the sample size, which needs to be calculated by “Cohen's d” (or others). In addition, the basic information about the subjects also needs to be added (Page 2, Line 58).

Results

In table 1, the numbers of the subjects were inconsistent. For example, the total number of students in 2nd was 27, while there were 26 male students and no female students. (There were also the same errors in BMI and Course) It’s essential to adjust the table to avoid the misunderstandings (Page 3, Line 126).

Discussion

The discussion quotes a lot of previous literatures but does not explain the reasons for the results. It is essential to add an explanation. The number of the reference was not adequate.

Author Response

Comments and Suggestions for Authors

This paper investigated the impact of COVID‐19 on physical activity and lifestyles in post‐confinement sports science undergraduates. The authors investigated the subjects' physical activity, diet, alcohol, and drug consumption habits using questionnaires and compared the differences between the infected and non-infected subjects. This paper is greatly weakened by the fact that there is no in-depth analysis of the reasons in the discussion section. In addition, the introduction part needs to be better organized.

Thank you for all suggestions, we answer one by one:

Abstract

The abstract does not correctly address all sections of the work. Conclusions are needed in this part.

Conclusions have been added. Our abstract contains: objective, methodology, results and conclusions.

Introduction

The introduction mainly described the harm of COVID-19 to society and humans, which didn’t focus on the highlights of the study. The author needs to add some information about the physical activity, diet, alcohol, and drug consumption habits of the patients. In addition, the number of relative researches was large, it would be better to add some literatures to support the summary of the previous studies.

Thank you for the comment. We have increased the information focusing on the highlights of the study. We have completed the information regarding to physical activity, diet, alcohol and drugs consumption in the context of Covid-19 lockdown. The information about our patients´habits are included in the Results section of the revised manuscript.

Methods

Lacking the description of the sample size, which needs to be calculated by “Cohen's d” (or others). In addition, the basic information about the subjects also needs to be added (Page 2, Line 58).

Thank you for the comment. We have added this information in the Method section of the reviewed manuscript, highlighted in yellow

Results

In table 1, the numbers of the subjects were inconsistent. For example, the total number of students in 2nd was 27, while there were 26 male students and no female students. (There were also the same errors in BMI and Course) It’s essential to adjust the table to avoid the misunderstandings (Page 3, Line 126).

Sorry for the mistake. We have included the correct information in the Results section of the revised manuscript, highlighted in yellow. We have revised all Tables and Figures

Discussion

The discussion quotes a lot of previous literatures but does not explain the reasons for the results. It is essential to add an explanation. The number of the reference was not adequate.This paper investigated the impact of COVID‐19 on physical activity and lifestyles in post‐confinement sports science undergraduates. The authors investigated the subjects' physical activity, diet, alcohol, and drug consumption habits using questionnaires and compared the differences between the infected and non-infected subjects. This paper is greatly weakened by the fact that there is no in-depth analysis of the reasons in the discussion section. In addition, the introduction part needs to be better organized.

Thank you for the comment. We have completely revised the discussion, and included additional information. We have amended in the reviewed manuscript, highlighted in yellow

Reviewer 3 Report

1. what you mean by moderate and severe obesity?

2.table 1: have you spend covid ? you mean got infected ?

3. table 3 : gis , you mean significant ?

4. table 4 : put that this value is for p value 

Author Response

Comments and Suggestions for Authors

  1. what you mean by moderate and severe obesity?

The difference between then is obtained following the classification wich has been used according to the values obtained in the BMI:

Underweight  (<18.5)

Normal weight (18.5-24.9)

Overweight (25-29.9)

Moderate Obesity (30-34.9)

Severe Obesity (>35)

It has been added in table 1. More information in:  http://dx.doi.org/10.4067/S0034-98872005000600015

2.table 1: have you spend covid ? you mean got infected ?

Yes, thank you for the observation. It has been modified.

  1. table 3 : gis , you mean significant ?

Yes, thank you. It’s a mistake.

  1. table 4 : put that this value is for p value 

Added in the title of the table. Thank you so much.

Round 2

Reviewer 2 Report

Thank you for responding to the proposed changes. I accept the changes made and the work for publication.